# SelectMDx and Multiparametric Magnetic Resonance Imaging of the Prostate for Men Undergoing Primary Prostate Biopsy: A Prospective Assessment in a Multi-Institutional Study

**DOI:** 10.3390/cancers13092047

**Published:** 2021-04-23

**Authors:** Martina Maggi, Francesco Del Giudice, Ugo G. Falagario, Andrea Cocci, Giorgio Ivan Russo, Marina Di Mauro, Giuseppe Salvatore Sepe, Fabio Galasso, Rosario Leonardi, Gabriele Iacona, Peter R. Carroll, Matthew R. Cooperberg, Angelo Porreca, Matteo Ferro, Giuseppe Lucarelli, Daniela Terracciano, Luigi Cormio, Giuseppe Carrieri, Ettore De Berardinis, Alessandro Sciarra, Gian Maria Busetto

**Affiliations:** 1Department of Urology, Sapienza Rome University, Policlinico Umberto I, 00161 Rome, Italy; martina.maggi@uniroma1.it (M.M.); francesco.delgiudice@uniroma1.it (F.D.G.); ettore.deberardinis@uniroma1.it (E.D.B.); alessandro.sciarra@uniroma1.it (A.S.); 2Department of Urology and Renal Transplantation, University of Foggia, Policlinico Riuniti, 71122 Foggia, Italy; ugo_falagario.559388@unifg.it (U.G.F.); luigi.cormio@unifg.it (L.C.); giuseppe.carrieri@unifg.it (G.C.); 3Department of Urology, University of Florence, Careggi Hospital, 50139 Florence, Italy; andrea.cocci@unifi.it; 4Department of Urology, University of Catania, 95100 Catania, Italy; giorgioivan.russo@unict.it (G.I.R.); marina.dimauro@unict.it (M.D.M.); 5Department of Urology, Padre Pio Clinic, 81034 Mondragone, Italy; giuseppe.sepe@clinicapadrepio.it; 6Department of Urology, Eboli Hospital, 84025 Eboli, Italy; fabio.galasso@eboliurologia.it; 7Department of Urology, Musumeci GECAS Clinic, 95030 Gravina di Catania, Italy; leonardi.r@tiscali.it; 8Department of Urology, Centro Uro-Andrologico (C.Ur.A.), 95024 Acireale, Italy; gabriele.iacona@gmail.com; 9Department of Urology, UCSF Helen Diller Comprehensive Cancer Center, University of California, San Francisco, CA 94143, USA; peter.carroll@ucsf.edu (P.R.C.); matthew.cooperberg@ucsf.edu (M.R.C.); 10Department of Urology, Veneto Institute of Oncology (IOV), 35128 Padua, Italy; angeloporreca@gmail.com; 11Department of Urology, European Institute of Oncology (IEO) IRCCS, 20141 Milan, Italy; matteo.ferro@ieo.it; 12Department of Emergency and Organ Transplantation-Urology, Andrology and Kidney Transplantation Unit, University of Bari, 74121 Bari, Italy; pappt1@yahoo.it; 13Department of Translational Medical Sciences, University of Naples Federico II, 80131 Naples, Italy; daniela.terracciano@unina.it

**Keywords:** prostate neoplasm, urinary biomarker, mpMRI, prostate biopsy, early detection, PSA

## Abstract

**Simple Summary:**

Prostate-specific antigen and digital rectal examination, used to guide prostate biopsy, often result in overdiagnosis of indolent prostate cancer (PCa) while missing clinically significant PCa (csPCa). The aim of this study was to evaluate the diagnostic accuracy of SelectMDx and its association with multiparametric magnetic resonance imaging (mpMRI) in predicting PCa in prostate biopsies. SelectMDx was revealed to be a good predictor of PCa, while with regards to csPCa detection, it was demonstrated to be less effective, showing results similar to mpMRI. The best diagnostic strategy to avoid unnecessary biopsy is performing SelectMDx after an initial negative mpMRI. Biopsy could be proposed for all cases of positive mpMRI and to those with a negative mpMRI followed by a positive SelectMDx.

**Abstract:**

Prostate-specific antigen (PSA) testing as the sole indication for prostate biopsy lacks specificity, resulting in overdiagnosis of indolent prostate cancer (PCa) and missing clinically significant PCa (csPCa). SelectMDx is a biomarker-based risk score to assess urinary HOXC6 and DLX1 mRNA expression combined with traditional clinical risk factors. The aim of this prospective multi-institutional study was to evaluate the diagnostic accuracy of SelectMDx and its association with multiparametric magnetic resonance (mpMRI) when predicting PCa in prostate biopsies. Overall, 310 consecutive subjects were included. All patients underwent mpMRI and SelectMDx prior to prostate biopsy. SelectMDx and mpMRI showed sensitivity and specificity of 86.5% vs. 51.9%, and 73.8% vs. 88.3%, respectively, in predicting PCa at biopsy, and 87.1% vs. 61.3%, and 63.7% vs. 83.9%, respectively, in predicting csPCa at biopsy. SelectMDx was revealed to be a good predictor of PCa, while with regards to csPCa detection, it was demonstrated to be less effective, showing results similar to mpMRI. With analysis of strategies assessed to define the best diagnostic strategy to avoid unnecessary biopsy, SelectMDx appeared to be a reliable pathway after an initial negative mpMRI. Thus, biopsy could be proposed for all cases of mpMRI PI-RADS 4–5 score, and to those with Prostate Imaging-Reporting and Data System (PI-RADS) 1–3 score followed by a positive SelectMDx.

## 1. Introduction

The European Association of Urology (EAU) guidelines on prostate cancer (PCa) recommend an individualized risk-adapted strategy for early detection of PCa, offering a prostate-specific antigen (PSA) test and digital rectal examination (DRE) to a well-informed man, aware of the related potential risks and benefits with at least ten to fifteen years of life expectancy [1]. However, use of PSA testing as the sole indication for prostate biopsy lacks specificity, resulting in overdiagnosis and potentially over-treatment of indolent PCa (i.e., non-aggressive), and, at the same time, may result in missing clinically significant PCa (csPCa) diagnoses in men with PSA levels below the cut-off value [2,3,4].

Many tools have been developed in recent years to avoid unnecessary biopsies and to overcome PSA limits. Improvements in risk stratification and a decrease of indolent PCa diagnosis have been reached, with the increased use of multiparametric magnetic resonance imaging (mpMRI) as one of the main diagnostic exams for PCa and as a tool for target biopsy [5,6].

To classify patients’ risk between low, intermediate or high (csPCa), in recent years different biomarkers and risk calculators (combining PSA and other risk factors), have been developed; however, their clinical benefit still needs to be proven [1]. Their development is guided by an appropriate risk stratification in order to avoid unnecessary biopsies and over-treatment for low-risk patients and to plan the correct treatment strategy for csPCa [7,8].

To consider initial prostate biopsy, blood biomarkers PHI and 4K Score, as well as urine biomarkers PCA3, SelectMDx and ExoDx have been developed [9,10,11,12,13,14,15,16,17,18]. To consider the treatment in patients with confirmed PCa, OncotypeDx, Prolaris and Decipher are available [19,20,21,22,23,24,25]. In order to evaluate a re-biopsy after an initially negative one, 4K score, PCA3, ExoDx and ConfirmMDx can help in the decision-making process [26,27,28,29,30,31,32,33,34].

SelectMDx is a novel biomarker-based risk score for assessing urinary *HOXC6* and *DLX1* mRNA expression combined with traditional clinical risk factors. Although SelectMDx is available in clinical practice to improve patient selection for initial prostate biopsy, previous studies have shown its ability to reduce the number of unnecessary biopsies, with current data being too limited to implement its use into routine screening programs [1]. Large prospective studies of the available biomarkers are still needed to assess whether their implementation in clinical practice is useful in providing guidance on decision making.

Following a prior positive single-institutional experience with SelectMDx test [17], we assessed a prospective and multi-institutional study with the aim to evaluate in a large cohort, if SelectMDx, associated with mpMRI, is a reliable method to predict PCa and csPCa for patients undergoing prostate biopsy.

## 2. Material and Methods

### 2.1. Study Population

In this multi-institutional prospective study, men from five different sites in Italy were consecutively enrolled between March 2018 and September 2019. All patients were scheduled for first prostate biopsy, and inclusion criteria were: elevated total PSA level (>3 ng/mL confirmed) and/or abnormal DRE. Exclusion criteria were those with history of PCa or different neoplasm under treatment, any medical treatment that could alter PSA value, any invasive treatments for BPH or any prior prostatic biopsy. Patient characteristics are shown in Appendix A.

### 2.2. mpMRI

All patients underwent mpMRI prior to biopsy using a 1.5 or 3.0 Tesla scanner (Achieva XR, Philips Medical System, Best, Netherlands; GE Discovery MR750, GE Healthcare, Chicago, IL, USA) with or without an endorectal coil. The functional technique of MRI was based on a combination of T2-weighted (T2W) images, diffusion-weighted imaging (DWI) and dynamic contrast-enhanced (DCE) studies. Lesions were characterized and graded using the Prostate Imaging-Reporting and Data System (PI-RADS) version 2.0 or 2.1, with a final grade from 1 to 5 indicating a greater probability of csPCa [35]. All mpMRI performed were analyzed by expert uro-radiologists (one to three per center, with minimum of five years of experience) blinded from patient characteristic, urine test score and biopsy outcome.

### 2.3. SelectMDx Sampling

Urine samples from the first-voided stream (approximately 30 mL) were collected after the DRE was conducted with a standard scheme of three strokes for each prostate lobe [36]. Samples have been shipped at room temperature to the central laboratory (MDxHealth Servicelab B.V., Nijmegen, The Netherlands) and stored at −80 °C. The SelectMDx score (MDx Health) is obtained combining different levels of expression of HOXC6 and DLX1 with clinical risk factors (age, DRE, total PSA, prostate volume) in a logistic regression model [37]. The results of the test are given as percentage of probability of positive or negative prostate biopsy for PCa (with two different probabilities for PCa and csPCa). Looking at previous experiences, in which a cut-off point was offered, here, only the probability is given [38].

### 2.4. Prostatic Biopsy

The trans-rectal ultrasound (TRUS) guided prostate biopsies have been performed by a single uro-radiologist for every center, all with more than 20 years of experience. The standardized biopsy scheme was: 12 random systematic cores from the peripheral zone of the prostate at the base, mid gland, and apex. In all patients where a mpMRI PI-RADS 3–5 lesion has been described, additional targeted samples (2 to 3 biopsy cores per lesion) have been obtained using an imaging fusion technique. All samples have been evaluated by experienced genitourinary pathologists and histological grading have been assessed following the Gleason grading system and Gleason Grade Groups (International Society of Urological Pathologist (ISUP) 2014) [39]. A csPCa was defined as ISUP score ≥2 (Gleason score ≥7) [1].

### 2.5. Statistical Analysis

Descriptive statistics were used to compare patients’ characteristics. A Pearson Chi-Square test was used to test association between categorical variables. Non-parametric Mann-Whitney U tests were used for comparisons of continuous covariates among groups. Sensitivity, specificity, and areas under the curves (AUC) were evaluated by computing receiver operating characteristic (ROC) curves. To compare the clinical utility of each tool alone or in association decision curve analysis were used (DCA). The SelectMDx scores were divided between positive (with the percentage of probability for PCa and csPCa) or negative for the suspicious of PCa by manufacturers’ report. PI-RADS 4–5 for mpMRI were considered positive while PI-RADS 1–3 negative. SelectMDx and positive mpMRI were used to evaluate the performance of tests together. To evaluate discordant cases, we carried out a simulated analysis, determining the number of avoided biopsies and missed PCa by SelectMDx results and PSAD values. Cut-off levels were >3 ng/mL for PSA, and ≥0.15 ng/mL/mL for PSAD.

Additionally, a multivariable stepwise logistic regression model (forward selection) was generated to assess the relative influence of those predictive variables that were significant upon univariate analysis on both the outcome of PCa and csPCa detection. Entering and removing limits were set at *p =* 0.05 and *p* = 0.10, respectively. In particular, age (continuous; <65 vs. ≥65 years), prostate volume (continuous; <56 vs. ≥56 mL), PCa familiarity (no vs. yes), DRE (negative vs. suspicious), total PSA (continuous, ng/mL), PSAD (<0.15 vs. ≥0.15), mpMRI score (negative 1–3 vs. positive 4–5) and SelectMDx (negative vs. positive) were included into the model. Finally, a locally weighted scatter plot smoother (LOWESS) function was used to graphically depict the relationship between the predicted probability of PCa or csPCa and the SelectMDx score provided within the ‘manufacturers’ report. Statistical analysis was performed using Stata (version 16.1) and SPSS (version 21.0) statistical programs, having set *p*-values <0.05 as statistically significant.

## 3. Results

Overall, 310 consecutive subjects have been included in our prospective analysis. Appendix A reports patients characteristics. SelectMDx was positive in 144 (46.5%) cases and a PI-RADS score 4–5 in 78 (25.2%) cases. A concordance between SelectMDx and mpMRI was found in 63.3% of cases. Out of 104 PCa (33.5%) detected at biopsy, 62 (20.0%) were csPCa. Table 1 reports stratification of subjects according to pathologic results at biopsy (i.e., PCa negative, all PCa, and csPCa). There was a significantly difference between PSA levels between the groups (*p* < 0.0001, and *p* = 0.001) (Table 1 and Appendix A).

SelectMDx score was positive in 86.5% and 87.1% of PCa and csPCa, respectively, and in 26.2% of cases with no PCa at biopsy (Table 1 and Figure 1a). The probability for a csPCa at the SelectMDx score was higher in csPCa (27.7 ± 18.1) than in PCa cases (25.9 ± 18.2) at biopsy (Table 1).

### 3.1. SelectMDx Performance for PCa and csPCa at Biopsy

The performance of SelectMDx compared to that of mpMRI PI-RADS score, and of the association between mpMRI and SelectMDx, PSA or PSAD to predict PCa and csPCa at biopsy is reported in Table 2.

SelectMDx and mpMRI PI-RADS scores showed sensitivity and specificity of 86.5% (95% CI 78.5–91.9) vs. 51.9% (95% CI 42.4–61.3), and 73.8% (95% CI 64.7–79.3) vs. 88.3% (95% CI 83.2–92.1), respectively, in predicting PCa at biopsy, and 87.1% (95% CI 76.2–93.5) vs. 61.3% (95% CI 48.8–72.4), and 63.7% (95% CI 57.5–69.4) vs. 83.9% (95% CI 78.7–87.9), respectively, in predicting csPCa at biopsy. Negative predictive value (NPV) and positive predictive value (PPV) were 91.6% vs. 78.4%, and 62.5% vs. 69.2%, respectively, for PCa and 95.2% vs. 89.7%, and 37.5% vs. 48.7%, respectively, for csPCa (Table 2). mpMRI sensitivity and specificity-considering positive those cases with PI-RADS score 3–5-were 82.7% (95% CI 74.2–88.8) and 77.7% (95% CI 71.5–82.8), respectively.

Sensitivity and specificity for both positive tests were 46.2% (95% CI 36.9–55.7) and 97.1% (95% CI 93.6–98.8), respectively, in predicting PCa at biopsy, and 54.8% (95% CI 42.5–66.6) and 91.9% (95% CI 87.8–94.8), respectively, in predicting csPCa at biopsy. Compared to the association of mpMRI and SelectMDx, that of mpMRI PI-RADS score and PSA showed slightly higher sensitivity and lower specificity in predicting both PCa and csPCa at biopsy, while the association of mpMRI PI-RADS score and PSAD showed lower sensitivity and specificity in predicting both PCa and csPCa at biopsy (Table 2).

SelectMDx score performance in predicting PCa and csPCa at biopsy was evaluated as area under the curve (AUC) of the receiver operating characteristics (ROC) in Figure 2. The AUC was 0.80 (95% CI 0.76–0.85) for PCa and 0.75 (95% CI 0.70–0.81) for csPCa; mpMRI PI-RADS score AUC was 0.70 (95% CI 0.65–0.75) for PCa and 0.73 (95% CI 0.66–0.79) for csPCa (Figure 2a,b). The AUC of using mpMRI and SelectMDx test together was 0.72 (95% CI 0.67–0.77) to detect PCa and 0.73 (95% CI 0.67–0.80) for csPCa, compared to 0.70 (95% CI 0.65–0.75) and 0.72 (95% CI 0.65–0.78), respectively, for the association mpMRI and PSA, and 0.67 (95% CI 0.62–0.71) and 0.68 (95% CI 0.62–0.75), respectively, for the association mpMRI and PSAD (Appendix A).

Decision curve analyses (DCA) were depicted to evaluate the clinical net benefit of SelectMDx and mpMRI (alone or in combination). The best combination for the detection of PCa and csPCa, found at DCA, was SelectMDX + mpMRI if compared to the association of mpMRI with other diagnostic tools (i.e., PSA and PSAD) (Figure 3a,b).

Moreover, at multivariable logistic regression analysis, mpMRI and SelectMDx were confirmed to be independently and strongly associated with both the outcomes of PCa and csPCa diagnosis at prostate biopsy (Odds Ratio [OR] PCa: 7.14, 95% CI: 3.31–15.39 and 25.57, 95% CI: 11.05–59.16 respectively; OR csPCa, 6.10, 95% CI: 2.94–12.64 and 9.49, 95% CI: 4.01–22.49, respectively; Appendix A). Finally, at LOWESS function analysis, the SelectMDx score given by the ‘manufacturers’ report (percentage (%) indicating the probability of PCa and/or csPCa from the test) showed an almost linear raising correlation when plotted against the multivariable adjusted predicted probability for the prostate biopsy detection of PCa (Figure 4a) and csPCa respectively (Figure 4b).

### 3.2. Impact of Implementation of SelectMDx Versus PSAD into the mpMRI Pathway to Select Patients Candidate for Prostate Biopsy

The distribution of SelectMDx scores according to PI-RADS findings in mpMRI is reported in Figure 1b and compared to those for total PSA in Appendix A. Interestingly, SelectMDx positivity increased from 30.3% for PI-RADS 1–2 cases, to 66.7% and 69.2% for PI-RADS 3 and 4–5 cases (*p* < 0.01); additionally, total PSA values were differently distributed among PI-RADS score groups (*p* < 0.01) (Figure 1b and Appendix A). With regards to PI-RADS 3 lesions, 59.3% (32/54) showed PCa at biopsy, and 14 (25.9%) were csPCa; SelectMDx score was positive in 81.3% of PI-RADS score 3 associated with PCa diagnosis and positive in 45.4% of those negative for PCa at biopsy. In PI-RADS 4–5 lesions, SelectMDx score was positive in 88.9% of PCa cases, and in 25.0% of those with no PCa at biopsy.

Cases with discordant tests were investigated to analyze the potential added value of implementing SelectMDx in the mpMRI diagnostic pathway. If we look at PI-RADS 1–2 cases, according to SelectMDx results, 30.3% (54/178) of patients would undergo biopsy with the detection of 16 (15.4%) PCa and 8 (12.9%) csPCa. Avoiding biopsy in patients with a PI-RADS score 4–5 and a negative SelecMDx test would result in 24 (30.8%) being spared biopsies within this category, while missing 6 (5.8%) PCa and 4 (6.5%) csPC. Regarding PI-RADS 3 cases, performing prostate biopsy only in those with a positive SelectMDx would result in 81.3% (26/32) of PCa detected, while avoiding biopsy in those with a negative SelectMDx would result in 18.8% (6/32) of PCa and 14.3% (2/14) of csPCa missed.

Performing prostate biopsy in patients with a PI-RADS score 1–2 and PSAD ≥ 0.15, would result in 48/178 (27.0%) biopsies performed in this category, with the detection of 6 (5.8%) PCa and 4 (6.5%) csPC. Avoiding biopsy in patients with a PI-RADS score 4–5 and PSAD < 0.15, would result in 28 (35.9%) spared biopsies within this category, yet missing 14 (13.5%) PCa and 10 (16.1%) csPC. If we perform prostate biopsy among PI-RADS score 3 cases only in those with PSAD ≥ 0.15, this would result in the detection of 8/32 (25.0%) PCa and 4/14 (28.6%) csPCa, while avoiding biopsy in those with PSAD < 0.15 would miss 24/32 (75.5%) PCa and 10/14 (71.4%) csPCa diagnosed within this category.

### 3.3. Impact of Different Screening Strategies to Select Patients Candidate for Prostate Biopsy

Several strategies of combining and sequencing SelectMDx and mpMRI have been simulated to investigate their impact in terms of number of avoided biopsies, missed PCa and csPCa (Table 3). Limiting biopsy to men with a positive SelectMDx would result in avoiding 53.5% (166/310) of biopsies, while missing 13.5% (14/104) of PCa and 12.9% (8/62) of csPCa; performing a biopsy only in those men with a positive mpMRI (PI-RADS 4–5) would avoid 74.8% (232/310) of biopsies and miss 48.1% (50/104) of PCa and 38.7% (24/62) of csPCa. Initial SelectMDx test followed by mpMRI if the test was positive and a subsequent biopsy if the mpMRI showed PI-RADS 4–5 findings, would result in 82.6% (256/310) of biopsies avoided, yet with 53.9% (116/104) of PCa and 45.2% (28/62) of csPCa missed. Initial mpMRI followed by biopsy for positive mpMRI cases (PI-RADS 4–5) and negative mpMRI cases (PI-RADS 1–3) (only if SelectMDx was positive), would result in avoiding 45.8% (142/310) of biopsies, while only missing 7.7% (8/104) of PCa and 6.5% (6/62) of csPCa.

## 4. Discussion

In order to tailor risk stratification and improve PCa detection, and particularly to reduce unnecessary prostate biopsies and diagnosis of indolent PCa, several tissues based urine and blood tests have been introduced to overcome PSA’s well-known limits [7,40,41,42,43,44,45,46,47]. Avoiding unnecessary prostate biopsies would allow for a reduction in the risk of side effects inherent to prostate biopsies, as well as associated health costs. This is especially relevant when a trans-rectal (TR) route is chosen, since evidence has suggested significantly higher infectious complications following TR biopsies, compared to a transperineal approach [48,49]. Among these tests, a 3-gene urinary panel using *HOXC6, TRD1* and *DLX1* were proposed in 2015 by Leyten et al. [50]. One year later, Van Neste et al. [38] reported that *TRD1* was not implementing the panel and that *HOXC6* and *DLX1* were sufficient for prediction of positive prostate biopsy and csPCa, with a sensitivity of 91%, specificity of 36% and NPV of 93%. Recently, a large multicenter trial including 1955 patients prior to initial prostate biopsy compared performance of urinary *HOXC6* and *DLX1* mRNA (combined with other risk factors) with Prostate Cancer Prevention Trial Risk Calculator (PCPTRC). AUC for molecular test was 0.85 and 0.76 for PCPTRC, demonstrating high sensitivity and NPV to detect csPCa [51]. The study by Rubio-Briones et al., analyzing 492 men with PSA 3–10 ng/mL, compared 2-gene urine-based molecular test targeting mRNAs with Prostate Cancer Antigen 3 (PCA3), European Randomized Screening in Prostate Cancer (ERSPC) and Prostate Biopsy Collaborative Group (PBCG) risk calculators. Focusing on patients with a Grading Group ≥2, the test avoided 37.2% of unnecessary biopsies, while delaying the diagnosis in 1.6% of cases between all patients. The authors concluded that the test could be useful to avoid unnecessary biopsies and to identify patients most likely to benefit from prostate biopsy [52].

To date, SelectMDx is not recommended by EAU guidelines since only few clinical trials prospectively investigated its performance in patients with an initial suspicious of PCa [1].

Initial prostate biopsy, in our prospective study, was scheduled for patients selected on the basis of PSA values or DRE results, and SelectMDx score was positive in 86.5% of PCa, 87.1% of csPCa, and in 26.2% of cases with no PCa at biopsy. Compared to mpMRI, SelectMDx had the best performance in predicting PCa and csPCa after biopsy. The association of mpMRI and SelectMDx compared to the association of mpMRI and other tools (i.e., PSA and PSAD), additionally, had the best performance. Moreover, SelectMDx showed a significant association with mpMRI results in terms of PI-RADS score, as test positivity significantly increased according to PI-RADS score (*p* < 0.001). It should be noted, however, that mpMRI results are particularly affected by the strategy used with PI-RADS 3 lesions (if considered as positive or negative cases). Indeed, considering PI-RADS 3 score as a positive test, with respect to PCa outcome, mpMRI sensitivity and specificity increased from 51.9% (if considered as a negative test) to 82.7%, and decreased from 88.3% (if considered as a negative test) to 77.7%, respectively.

In our personal clinical experience SelectMDx demonstrated to be a good predictor of PCa. We reported that in patients before initial biopsy it could reach high levels of sensitivity and specificity for the diagnosis of all PCa (AUC 0.80), while it seems slightly less effective in detecting csPCa (AUC 0.75). Moreover, with regards to csPCa detection, SelectMDx results were similar to mpMRI. On the contrary, mpMRI demonstrated a better diagnostic performance with respect to csPCa outcome (AUC 0.73), than to PCa outcome (AUC 0.70).

According to EAU guidelines on PCa, mpMRI is recommended before performing prostate biopsy in biopsy-naïve patients with clinical suspicion of PCa [1]. In the era of mpMRI pathway, to aid clinicians in decision making (i.e., to decide whether a prostate biopsy can be omitted, in an effort to avoid unnecessary biopsy), and to improve the detection of csPCa while limiting the detection of indolent cases, combining mpMRI with serum biomarkers would be of clinical value, yet the optimal sequence and timing remains to be determined. According to our simulated strategies, limiting biopsy to men with a mpMRI PI-RADS score 4–5 would have resulted in avoiding 74.8% (232/310) of biopsies, yet missing 38.7% (24/62) of csPCa. Interestingly, implementing SelectMDx test after a mpMRI PI-RADS score 1–3, and performing a biopsy in mpMRI PI-RADS score 4–5 and in those with PI-RADS score 1–3 yet with a positive SelectMDx, would have resulted in missing only 6.5% (6/62) of csPCa, still avoiding 45.8% (142/310) of biopsies. SelectMDx could potentially lower the number of mpMRI scans and biopsies performed without increasing the risk of missing csPCa. However, our results suggest that upfront SelectMDx, although in 82.6% of avoided biopsies, is associated with a high risk of missing csPCa (45.2%).

Cases with discordant tests were analyzed to better investigate the potential added value of implementing SelectMDx in the mpMRI diagnostic pathway. With regards to PI-RADS 3 lesions, which are equivocal by nature, several possible clinical factors have been evaluated as predictors of positive biopsy [53]. In this context, SelectMDx might aid the decision-making scenario (i.e., whether a prostate biopsy versus observation should be advised). In our experience, performing a biopsy only in those with a positive SelectMDx, would result in the detection of 81.3% of PCa and 85.7% of csPCa diagnosed within this category. Differing from data reported by some authors, in our analysis, PSAD with a cut-off value of ≥0.15 to decide on the need for biopsy, did not show a clinical benefit for the detection of PCa and csPCa.

In conclusion, our analysis suggests several points of interest: (I) SelectMDx demonstrated a valid diagnostic accuracy for the detection of PCa. (II) With regards to csPCa detection, SelectMDx showed a reliable diagnostic performance, and comparable to that of mpMRI. (III) In the era of mpMRI pathway, the association of mpMRI and SelectMDx showed the best performance, compared to the association of mpMRI and other tools (i.e., PSA and PSAD). (IV) Upfront mpMRI followed by SelectMDx in cases with PI-RADS 1–3 scores to decide the need to undergo biopsy, appeared a reliable strategy to avoid unnecessary biopsies, with a reasonable csPCa detection rate. (V) Regarding the management of equivocal PI-RADS 3 lesions, SelectMDx performed better than PSAD in selecting patients for biopsy.

Our study warrants certain limitations. Due to the multicentric nature of our study, diagnostic performance data for mpMRI could be at least partially affected by variability in mpMRI-related factors (i.e., inter-reader variability, readers experience and technical performance) among the different centers included in the study. In our study all mpMRI performed have been analyzed by expert uro-radiologists, 1 to 3 per center, with minimum 5 years of experience, adopting high standards of image quality. However, a central review was not assessed. Moreover, we did not perform a cost-effectiveness analysis of SelectMDx compared to that of mpMRI, which is a main issue that should be covered before its routinely implementation in clinical practice for an extended population. To this end, although elevated costs represent a main limit, recent studies suggest that saving healthcare costs for PCa diagnosis is possible and could be done because SelectMDx quality-adjusted life years (QALYs) increase. If comparing standard of care for PCa diagnosis with SelectMDx, it could be a potentially correct cost-effective strategy [54,55,56]. On the contrary, comparing the performance of mpMRI with biomarkers, it seems that minimizing cost and maximizing effectiveness, would be the optimal strategy [48]. Future research should address the cost-effectiveness of SelectMDx compared with mpMRI and other biomarkers in order to offer to the clinicians the best diagnostic toll at the lowest price.

## 5. Conclusions

In our clinical experience, SelectMDx was revealed to be a good predictor of PCa, while, with regards to csPCa detection, it was demonstrated to be less effective, showing results similar to mpMRI. With analysis of strategies assessed to define the best diagnostic strategy to avoid unnecessary biopsy without missing csPCa, SelectMDx appeared after an initial negative mpMRI to be a reliable pathway. Thus, biopsy could be proposed to all cases of mpMRI PI-RADS 4–5 score, and to those with mpMRI PI-RADS 1–3 score, followed by a positive SelectMDx.

## Figures and Tables

**Figure 1 cancers-13-02047-f001:**
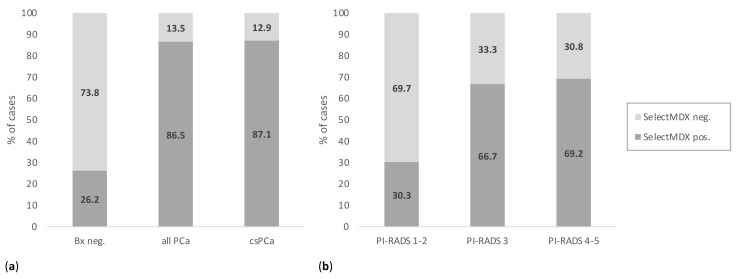
SelectMDx positive and negative results according to (**a**) histologic diagnosis for PCa at biopsy; and (**b**) to PI-RADS score at mpMRI. (PCa = prostate cancer; PI-RADS = Prostate Imaging-Reporting and Data System; mpMRI = multiparametric magnetic resonance imaging; Bx = biopsy; csPCa = clinically significant PCa).

**Figure 2 cancers-13-02047-f002:**
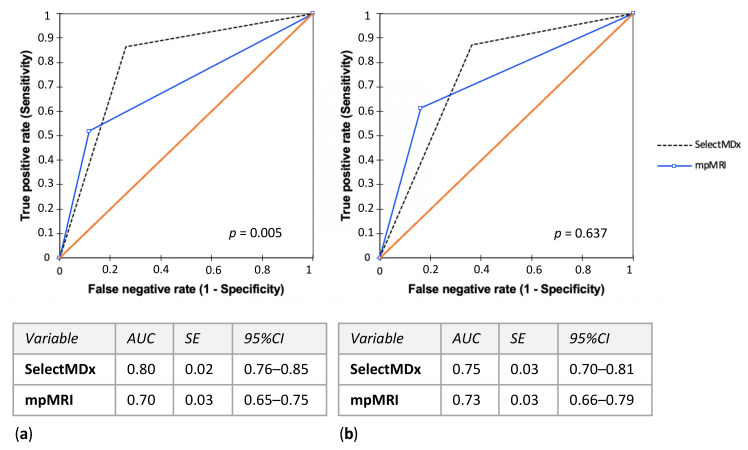
SelectMDx score and mpMRI PI-RADS score performance evaluated as area under the curve (AUC) of the receiver operating characteristics (ROC) in predicting (**a**) PCa and (**b**) csPCa histological diagnosis at biopsy. (mpMRI = multiparametric magnetic resonance imaging; PI-RADS = Prostate Imaging-Reporting and Data System; PCa = prostate cancer; csPCa = clinically significant PCa; AUC = area under the curve; SE = standard error; CI = confidence interval).

**Figure 3 cancers-13-02047-f003:**
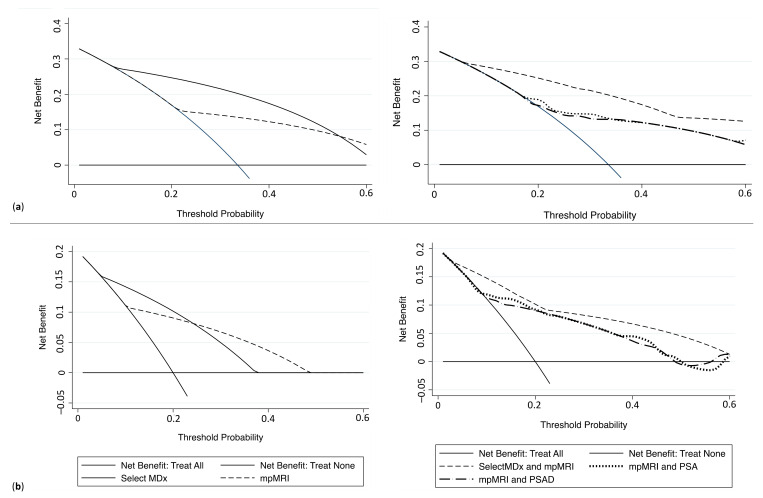
Decision curve analysis comparing clinical utility of SelectMDx score, mpMRI, and the associations mpMRI+ SelectMDx, mpMRI + PSA and mpMRI + PSAD for detecting (**a**) PCa and (**b**) csPCa. (mpMRI = multiparametric magnetic resonance imaging; PSA = prostatic-specific antigen; PCa = prostate cancer; csPCa = clinically significant PCa).

**Figure 4 cancers-13-02047-f004:**
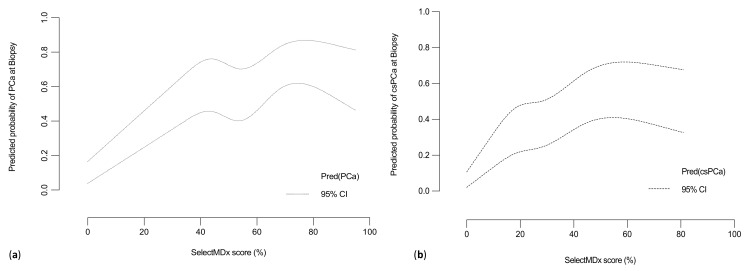
Multivariable adjusted Locally Weighted Scatter Plot Smoother (LOWESS) function depicting the predicted probability of (**a**) PCa or (**b**) csPCa detection rate and SelectMDx score (%). (PCa = prostate cancer; csPCa = clinically significant PCa).

**Table 1 cancers-13-02047-t001:** Patient characteristics stratified according to prostatic biopsy results (number, %, mean ± SD, median, range).

Parameter	Negative PCa	All PCa	csPCa	*p*-Value *	*p*-Value **
Number of cases *n* (%)	206 (66.5)	104 (33.5)	62 (20.0)		
Age (years)					
mean ± SD	64 ± 7.7	64 ± 8.2	63 ± 9.0	0.691	0.518
median	65	65	64		
range	44–79	45–79	45–79		
Total PSA (ng/mL)					
Mean ± SD	6.9 ± 4.1	9.0 ± 4.3	9.5 ± 5.0	<0.0001	0.0001
median	6.1	8	8.1		
range	1.0–19.5	1.0–19.9	1.2–19.9		
PSAD (ng/mL/mL)					
Mean ± SD	0.13 ± 0.08	0.18 ± 0.11	0.18 ± 0.12	<0.0001	0.0001
median	0.1	0.15	0.16		
range	0.02–0.36	0.02–0.53	0.02–0.53		
SelectMDx score, n (%)					
negative	152 (73.8)	14 (13.5)	8 (12.9)	<0.0001	<0.0001
positive	54 (26.2)	90 (86.5)	54 (87.1)		
probability for csPCa (%)					
mean ± SD	6.9 ± 13.9	25.9 ± 18.2	27.7 ± 18.1	<0.0001	<0.0001
median (range)	0(0–62)	24.0(0–81)	25.0(0–81)		
mpMRI PI-RADS score, *n* (%)					
PI-RADS 1–2	160 (77.7)	18 (17.3)	10 (16.1)	<0.0001	<0.0001
PI-RADS 3	22 (10.7)	32 (30.8)	14 (22.6)		
PI-RADS 4–5	24 (11.6)	54 (51.9)	38 (61.3)		
SelectMDx score and					
mpMRI PI-RADS score, *n* (%)					
SelectMDx positive, mpMRI positive	6 (2.9)	48 (46.2)	34 (54.8)	0.001	<0.0001
SelectMDx negative, mpMRI negative	134 (65.0)	8 (7.7)	4 (6.5)		
SelectMDx positive, mpMRI negative	48 (23.4)	42 (40.4)	20 (32.2)	0.008	<0.0001
SelectMDx negative, mpMRI positive	18 (8.7)	6 (5.7)	4 (6.5)		

*n* = number, SD = standard deviation, PSA = prostate-specific antigen, PSAD = PSA density, mpMRI = Multiparametric magnetic resonance imaging, PI-RADS = Prostate Imaging-Reporting and Data System, PCa = prostate cancer, csPCa = clinically significant PCa. * *p*-values indicating the comparison between the Negative PCa vs. PCa group; ** *p*-values indicate the comparison between the negative PCa vs. csPCa group.

**Table 2 cancers-13-02047-t002:** Performance of individual parameters and their combination to predict PCa and csPCa on biopsy (% and 95% CI).

Parameter	All PCa				csPCa			
	Sensitivity	Specificity	NPV	PPV	Sensitivity	Specificity	NPV	PPV
SelectMDx	86.5 (78.5–91.9)	73.8 (64.7–79.3)	91.6	62.5	87.1 (76.2–93.5)	63.7 (57.5–69.4)	95.2	37.5
mpMRI PI-RADS	51.9 (42.4–61.3)	88.3 (83.2–92.1)	78.4	69.2	61.3 (48.8–72.4)	83.9 (78.7–87.9)	89.7	48.7
mpMRI PI-RADS and SelectMDx	46.2 (36.9–55.7)	97.1 (93.6–98.8)	78.1	88.9	54.8 (42.5–66.6)	91.9 (87.8–94.8)	89.1	63.0
mpMRI PI-RADS and total PSA	50.0 (40.6–59.4)	90.3 (85.4–93.7)	78.2	72.2	58.1 (45.7–69.5)	85.5 (80.5–89.3)	89.1	50.0
mpMRI PI-RADS and total PSAD	38.5 (29.7–48.1)	95.1 (91.1–97.4)	75.4	80.0	45.2 (33.4–57.5)	91.1 (86.9–94.1)	86.9	56.0

mpMRI = Multiparametric magnetic resonance imaging, PI-RADS = Prostate Imaging-Reporting and Data System, PSA = prostate-specific antigen, PSAD = PSA density, PCa = prostate cancer, csPCa = clinically significant PCa, NPV = negative predictive value, PPV = positive predictive value.

**Table 3 cancers-13-02047-t003:** Prostate cancer (PCa) and clinically significant prostate cancer (csPCa) detection rate, avoided biopsies and missed PC and csPC among the study population according to different strategies.

Strategy	Avoided Biopsies, *n* (%)	PCa, *n* (%)		csPCa, *n* (%)	
	*n*= 310	*n* = 104		*n* = 62	
		**Detected**	**Missed**	**Detected**	**Missed**
strategy 1	166 (53.5)	90 (86.5)	14 (13.5)	54 (87.1)	8 (12.9)
strategy 2	178 (57.4)	86 (82.7)	18 (17.3)	52 (83.9)	10 (16.1)
strategy 3	232 (74.8)	54 (51.9)	50 (48.1)	38 (61.3)	24 (38.7)
strategy 4	220 (71.0)	74 (71.1)	30 (28.9)	46 (74.2)	16 (25.8)
strategy 5	256 (82.6)	48 (46.1)	116 (53.9)	34 (54.8)	28 (45.2)
strategy 6	124 (40.0)	102 (98.1)	2 (1.9)	60 (96.8)	2 (3.2)
strategy 7	142 (45.8)	96 (92.3)	8 (7.7)	58 (93.5)	6 (6.5)

Strategy 1: get a SelectMDX test alone and biopsy any positive test; strategy 2: get a mpMRI alone and biopsy any positive mpMRI, considered as PI-RADS score 3–5; strategy 3: get a mpMRI alone and biopsy any positive mpMRI, considered as PI-RADS score 4–5; strategy 4: get a SelectMDx test first and if negative do not biopsy; if positive, get a mpMRI and do a biopsy only if it is + (considered as PI-RADS score 3–5); strategy 5: get a SelectMDx test first and if negative do not biopsy; if positive, get a mpMRI and do a biopsy only if it is + (considered as PI-RADS score 4–5); strategy 6: get a mpMRI first and if positive (considered as PI-RADS score 3–5), do a biopsy; if negative, get a SelectMDx and do a biopsy only if it is positive; strategy 7: get a mpMRI first and if positive (considered as PI-RADS score 4–5), do a biopsy; if negative, get a SelectMDx and do a biopsy only if it is positive.

## Data Availability

The data presented in this study are available on request from the corresponding author.

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
