# Peer review of "SelectMDx and Multiparametric Magnetic Resonance Imaging of the Prostate for Men Undergoing Primary Prostate Biopsy: A Prospective Assessment in a Multi-Institutional Study"

_cancers, 2021, doi:10.3390/cancers13092047_

Round 1

Reviewer 1 Report

This article overviews the potential clinical utility of SelectMDx as compared to mpMRI in the detection of any PCa and clinically significant PCa. Their selected patient cohort appropriately reflects patients that we would assume to have a reasonable chance of screen detected disease on biopsy.  The performance of SelectMDx is overall quite good however the authors' attempt to introduce it's value after MPMRI is not clearly shown other than in simulated scenarios. The authors may want to temper this statement a bit. The strength of this study is the use of both 12 core random biopsies and targeted biopsies in those patients with mpMRI abnormalities. Overall, I believe this is a high impact and clinically thought provoking study. However, further study is necessary before this can be adopted into clinical practice.

Results:

In section 2.1 there is a typo and it should read "There was no significant difference between PSA levels ..." Many authors would argue that the significant predictor in Table 1 is the PSA density as this has much higher predictive value for PCa shown in many studies and yours.

Appropriate. the authors could consider a comparison of MRI PIRADs 1-2 with PSAD>0.15 and the MRI PIRADs 1-2 with SelectMDx +ve patients. Of note the PSAD would result in 48 biopsies and 6 PCa diagnoses but with SelectMDx positive would result in 54 biopsies and 16 PCa diagnoses. it would be interesting to know the PSAD values and prostate volumes of all patients who were SelectMDx only detected. 

Discussion:

The authors have overviewed their results in the context of the known literature about SelectMDx. However, they need to be clearer about the benefits to avoiding biopsy in patients with classical indications for biopsy. There are several potential benefits with the greatest being decreasing overall risk of severe infection and further expansion on this is necessary. Once this is explained, the authors should acknowledge that other biopsy techniques such as transperineal approaches may be more comfortable and pose less risk to patients.

Reviewer 2 Report

The aim of the manuscript is to evaluate the diagnostic accuracy of SelectMDx and its association with mpMRI in predicting PCa on prostate biopsies. 310 subjects with SelectMDx and mpMRI prior to prostate biopsy were included. The current manuscript should be a follow-up study of the previous published study back in 2020 by Busetto et al. (https://doi.org/10.1007/s00345-020-03359-w), where 52 subjects were analysed, although there is no specific reference in the manuscript indicating this.

Major comments:

Line 30-36: In the “Simple Summary” the authors use many abbreviations without being explained in advanced. Unfortunately, the text is useless if the reader is not familiar with the abbreviations.

Line 86-88: To my understanding the present manuscript is based on previously published in 2020 study by Busetto et al. (https://doi.org/10.1007/s00345-020-03359-w). The authors in the previously published manuscript evaluated the diagnostic accuracy of SelectMDx and its association with mpMRI in predicting PCa and clinically significant PCa (csPCa) on prostate biopsies among men scheduled for initial prostate biopsy. That study included 52 men. The authors must acknowledge and highlight this fact in the introduction. Additionally, similarities or deviations must be discussed, as well.

Line 89: The manuscript must follow the structure: Introduction, Material and methods, Results, Discussion, Conclusions, and References. In the current form, the authors, after the “Introduction” section, jump into “Results”.

Line 90: The “Study population” sub-section belongs to “Material and methods” section.

Line 94-95: The sentence (“Between 104 detected PCa (33.5%) at biopsy, 62 (20.0%) were csPCa.”) does not make any sence. Please, rephrase.

Line 97: It is not clear which groups were compared. Is it all three together (Negative PCa, all PCa, csPSA)? Is it was a paired comparison among groups? Please, clarify. In addition, indicate which statistical test used for the comparison.

Line 128-129: In Table 2, the confidence intervals must be presented in percentage, as it was indicated in line 129.

Line 145: In the Figure 2a and 2b the 95% CI must be plotted, as well.

Line 252-253: Is the difference between the two AUC values statistical significant? Please, provide a p-value.

Line 352-355: The authors indicate all possible hypothesis tests (parametric and non-parametric) without giving specific information on the comparisons they performed. The authors must provide specific information indicating comparison cases and an appropriate hypothesis test.

Line 358-360: The authors must explain how they concluded to a percentage of probability for PCa and csPCA from a positive and negative SelectedMDx score.  Also why percentage of probability and not just a probability value?

Line 362: Were there any avoided biopsies within the cohort? I thought that all patients had a biopsy, right? It is not clear what you want to say.

Material and Methods: the authors must provide, at least in the supplementary material, a detailed information of all 310 patient along with the values of age, PSA, PSAD, SelectMDx score, Gleason score, PI-RADS score.

Statistical analysis: The statistical analysis is rather complicated. It is not clear which statistical comparisons took place. For the multiple-comparisons, a p-value correction must be applied. Additionally, a multivariate logistic regression model must be appropriate for such analysis. For the AUC comparisons, a p-value must be provided.

Minor comments:

Line 37: PSA abbreviation must be explained before use.

Line 45: csPSA abbreviation must be explained before use.

Line 57: “to a well-informed man”: The meaning in the current content is not clear.

Line 59: “indolent PCa” please replace with “indolent PCa (non-aggresive)” for clarity.

Line 67 - 69: Please provide references.

Line 67 - 69: Please replace “… between PCa and csPCa” with “… between low and intermediate or high (csPSA)”. Otherwise, the readers will get confused when reading lines 71-72.

Line 69: Typo error. “However” is used twice.

Line 73-77: The authors list a number of biomarkers. In the connection to the following paragraph, it is not clear why the authors selected to go for SelectMDx biomarker and why they do not consider the other ones.

Line 154: Please, increase the font size of the figure legends and axis labels. Currently, the fonts are too small. Also, use different line style and thicker drawing lines. Please, use colour-blind friendly schemes.

Round 2

Reviewer 2 Report

For supporting open science and following FAIR principles it would be nice if the authors share with us the "raw" data in an Excel or CSV (comma separated value) file.

No further comments from my side and many thanks to the authors for their clarifications.

This manuscript is a resubmission of an earlier submission. The following is a list of the peer review reports and author responses from that submission.